# Transient Receptor Potential Ankyrin 1 (TRPA1) Modulation by 4-Hydroxynonenal (4-HNE) in Pancreatic Adenocarcinoma Cell Lines: Putative Roles for Therapies

**DOI:** 10.3390/ph17030344

**Published:** 2024-03-06

**Authors:** Florentina Piciu, Dan Domocos, Gabriela Chiritoiu, Marioara Chiritoiu-Butnaru, Maria Mernea, Cezar Gabriel Popescu, Dragos Paul Mihai, Bianca Galateanu, Ariana Hudita, Alexandru Babes, Dana Cucu

**Affiliations:** 1Department of Anatomy, Animal Physiology and Biophysics, Faculty of Biology, University of Bucharest, Spl. Independentei 91-95, 050095 Bucharest, Romania; florentina.cojocaru@bio.unibuc.ro (F.P.); dan.domocos@gmail.com (D.D.); maria.mernea@bio.unibuc.ro (M.M.); cezar-gabriel.popescu@s.unibuc.ro (C.G.P.); alexandru.babes@bio.unibuc.ro (A.B.); 2Research Institute of the University of Bucharest (ICUB), University of Bucharest, 90-92 Sos. Panduri, 050663 Bucharest, Romania; 3Cell Signalling Research Group, Institute of Biochemistry of the Romanian Academy, Splaiul Independenţei 296, 060031 Bucharest, Romania; 4Department of Molecular and Cellular Biology, Institute of Biochemistry of the Romanian Academy, Splaiul Independentei 296, 060031 Bucharest, Romania; gabriela_chiritoiu@yahoo.com (G.C.); mari.chiritoiu@gmail.com (M.C.-B.); 5Faculty of Pharmacy, “Carol Davila” University of Medicine and Pharmacy, 6 Traian Vuia Street, 020956 Bucharest, Romania; dragos_mihai@umfcd.ro; 6Department of Biochemistry and Molecular Biology, Faculty of Biology, University of Bucharest, Spl. Independentei 91-95, 050095 Bucharest, Romania; bianca.galateanu@bio.unibuc.ro (B.G.); ariana.hudita@bio.unibuc.ro (A.H.)

**Keywords:** TRPA1, 4-HNE, PDAC, pancreatic adenocarcinoma

## Abstract

Background: Transient receptor potential channels (TRP) are overexpressed in some pancreatic adenocarcinoma (PDAC) patients and cell lines, settling them as putative therapeutic targets in this disease. Reactive oxygen species (ROS), with levels increased in PDAC, modulate some members of the TRP family renamed “redox channels”. Here, we investigate the direct effects of 4-hydroxinonenal (4-HNE) on TRPA1, natively expressed in PDAC cell lines and in association with cell migration and cell cycle progression. Methods: We performed microfluorimetry experiments, while the activation of resident membrane channels was investigated using confocal microscopy. We applied a prospective molecular docking of 4-HNE using Autodock and AutoDock Tools4. Also, we simulated the diffusion of 4-HNE through the membrane from the extracellular space with the Permeability of Molecules across Membranes (PerMM) web server. The analysis of cell migration was performed using the wound healing assay, and cell cycle progression was acquired using a Beckman Coulter CytoFlex flow cytometer. Results: Our results show, for the first time in PDAC, that 4-HNE diffuses through the cell membrane and rapidly activates Ca^2+^ uptake in PDAC cells. This process depends on TRPA1 activation, as 4-HNE forms a covalent binding with a pocket-like region within the intracellular N-terminal of the channel, shaped by the cysteine residues 621, 641, and 665. The activation of TRPA1 by 4-HNE inhibits cell migration and induces cell cycle arrest in the G2/M phase. Conclusions: Our study brings new insights into the effects of 4-HNE, highlighting the activation of the TRPA1 channel, a druggable, putative target for PDAC-expressing tumors.

## 1. Introduction

Pancreatic ductal adenocarcinoma (PDAC) represents one of the most aggressive forms of cancer in adults, with an incidence rate of 6.4 worldwide and 18.4 in Europe per 100,000 population, as reported for the year 2020 [1]. PDAC is a disease associated with a very poor prognosis and overall survival of <1 year, while in metastatic tumors, it is <6 months, using the current standard of care treatment. The limited treatment options for PDAC consist of surgery followed by chemo- and radiotherapy or adjuvant therapy with a combination of chemotherapeutic agents (FOLFIRINOX) and gemcitabine regimens [2]. Alternative therapies are currently under development but show suboptimal efficacy and induce drug resistance. Therefore, novel treatment options to control this devastating disease are urgently needed.

In the last decade, many groups have probed several genes or proteins to salvage the therapeutic limitations induced by the late detection of PDAC, when most patients are diagnosed with metastases. Among them are members of the superfamily of transient receptor potential channels (TRP), indicated as putative targets in those cases where overexpression is measurable [3]. Lately, new perspectives arose when some members of the TRP family were discovered to be reactive oxygen species (ROS) sensors. In this context, TRPC4, TRPC5, TRPV1, TRPV4, TRPA1, TRPM2, and TRPM7 were renamed “redox channels” [4]. Among them, TRPA1 is the most sensitive channel to ROS, being activated by extracellular hydrogen peroxide (H_2_O_2_), hydroxyl radical (HO·), nitric oxide, or endogenously generated 4-hydroxinonenal (4-HNE), which induce redox modification of the thiol groups. Thus, the high ROS sensitivity of TRPA1 is mediated by the intracellular cysteine residues (Cys621, Cys641, Cys665) and Lys708, localized on their N-terminal segments [5]. 

Intracellular ROS oxidize lipids, proteins, and DNA, and depending on the concentration, they also induce fine-tuned modulation of tumorigenesis [6]. For instance, lower levels of 4-HNE were noticed in cancer tissues, suggesting a potential anti-malignant effect of this compound [7]. Moreover, high levels of 4-HNE impaired mitochondrial function by calcium overload in chronic postischemic remodeling [8]. 

In PDAC cells and patients, TRPA1 is overexpressed [9], but so far, the effects of ROS on TRP channels in PDAC cells have not yet been reported. In this study, we propose to investigate the effect of 4-HNE on the TRPA1 channel and its relation to cell migration and cell cycle progression. We also aim to offer here a model of the interaction between TRPA1 and 4-HNE with the hope that this channel may be a promising target in therapies related to the increased 4-HNE production.

## 2. Results

Our previous studies showed that pancreatic adenocarcinoma cell lines express higher levels of the TRPA1 gene and protein than a non-tumoral pancreatic ductal cell line [10].

### 2.1. 4-HNE Directly Induces Ca^2+^ Uptake via TRPA1 Channel Activation

To determine whether the endogenously expressed TRPA1 channels respond to 4-HNE, as reported on HEK cells heterologously expressing the channel [11], we performed microfluorimetry experiments. 

Figure 1a illustrates Ca^2+^ imaging experiments on PANC-1 cells stimulated with 50 μM 4-HNE. All cells (*n* = 30) evoked robust Ca^2+^ transients when stimulated. All 4-HNE-activated cells also responded to the specific TRPA1 agonist allyl isothiocyanate (AITC, 100 μM for 1 min, Figure 1a). To confirm that Ca^2+^ enters the cells through TRPA1, in the extracellular solution (ES), we added the highly specific and potent TRPA1 antagonist A-967079 (10 μM) and applied it during the second 4-HNE challenge (Figure 1b, red traces). The compound inhibited the second response by 95.7% compared to the control response (*n* = 35). The normalized ΔF/F0 values significantly decreased from 0.47 ± 0.06 in control (*n* = 17) to 0.02 ± 0.003 in treated cells (*n* = 35) (*p* < 0.001, two-sample Student’s *t*-test; Figure 1c).

Next, we wanted to investigate whether 4-HNE modulates the TRPA1 function by activating resident membrane channels or enhancing TRPA1 trafficking to the plasma membrane. To address this hypothesis, we performed fluorescence microscopy experiments monitoring TRPA1 intracellular localization in the presence and absence of 50 μM 4-HNE. We used cells treated with 50 μM H_2_O_2_ as the positive control. To evaluate the potential effect of 4-HNE on the trafficking of TRPA1 to the plasma membrane, we assessed the colocalization of TRPA1 with calnexin, an ER resident protein widely accepted as an ER marker, and with actin, which partially decorates the plasma membrane as cortical actin (Figure 2a). 

Colocalization with plasma membrane actin or ER (calnexin) markers was evaluated using the overlap Manders’ coefficient applied to confocal images. The quantification of the Manders coefficient showed that in control conditions, the overlap of TRPA1/~calnexin was ~20% higher than for TRPA1/actin (Figure 2b). When we treated the cells with 50 μM 4-HNE for 24 h, the Manders’ coefficient for TRPA1/calnexin slightly augmented from 0.41 ± 0.09 to 0.48 ± 0.09 (*p* = 0.01, two-sample Student’s *t*-test, *n* = 30 cells, Figure 2b, red box chart). In contrast, for TRPA1/actin, the values did not change significantly (Figure 2c, red box chart). To compare the effect of 4-HNE with a different ROS product, we treated the cells with hydrogen peroxide (H_2_O_2_). As depicted in Figure 2b, blue box charts, in 50 μM H_2_O_2_ for 24 h, the Manders’ coefficient for TRPA1/actin significantly increased by 37.9% (*p* = 0.002, two-sample Student’s *t*-test, *n* = 30 cells). In contrast, colocalization of TRPA1 with calnexin decreased by 21.8% (*p* = 0.001, two-sample Student’s *t*-test).

### 2.2. 4-HNE Stabilizes the Open Structure of TRPA1

As proven above, 4-HNE most likely directly activates TRPA1 at a site localized within the channel. 4-HNE forms protein adducts by covalent binding, reacting with the cysteine residues [12]. The residue Cys621 along with Cys641 and Cys665 are critical for the electrophilic activation and stabilize an open structure of the channel, forming a pocket-like region [13]. For simulations, we used the AlphaFold model, which comprises the structure of the full-length ankyrin repeat domain of TRPA1 predicted with high accuracy (pLDDT > 90) and with high confidence (90 > pLDDT > 70). We initially performed a prospective docking of 4-HNE in the broader region centered into the cysteine pocket (Figure 3a).

Our results show that 4-HNE has an affinity for that site, the best pose presenting an estimated free energy of binding of −4.49 kcal/mol (Figure 3b). We measured the distance between the atoms of 4-HNE and Cys621 putatively involved in covalent bonding and found it to be 4.4 Å. We searched for the best docking pose with the shortest distance between these atoms and identified the distance of 3.7 Å for the pose illustrated in Figure 3c. In this case, the estimated free energy of binding was −4.35 kcal/mol. The location of 4-HNE in this pose was used as the input for the covalent docking of 4-HNE into the binding site. A covalent bond was then modeled between the atoms of 4-HNE and Cys621. Figure 3d illustrates the best covalent docking pose, and Figure 3e shows the residues involved in non-covalent interactions with 4-HNE.

### 2.3. Exogenous 4-HNE Passively Diffuses across the Cell Membrane

Next, we questioned whether exogenous 4-HNE and other similar lipid peroxidation products (4-ONE and 4-HHE) penetrate the cell membrane and consecutively bind covalently to the intracellular Cys residues. To this end, we simulated the diffusion across the lipid bilayer with the PerMM web server. We used cinnamaldehyde and citric acid as positive and negative controls, respectively.

Considering the predicted membrane transfer energy (ΔGt_ransf_) profiles along the lipid bilayer standard (Z), the transport of citric acid across the cell membrane requires high transfer energies (>12 kcal/mol). Accordingly, passive diffusion is highly unlikely to occur (Figure 4).

From the investigated peroxidized lipids, 4-HNE and 4-ONE had comparable transbilayer energy profiles, with slightly lower transfer energies for 4-ONE across the hydrophobic portion of the membrane (±15 Å from the bilayer center) and an overall more symmetrical profile. However, 4-HNE required higher transfer energies to diffuse (±15–30 Å from the bilayer center) across the lipid acyl chain region. Moreover, 4-HNE showed the second lowest membrane binding energy (−3.62 kcal/mol) and the second highest predicted logPerm (−0.81) among the assessed lipid aldehydes. We noticed that cinnamaldehyde required the lowest transfer energy among the investigated compounds, with a maximum energy of approximately −2 kcal/mol. Cinnamaldehyde had the highest affinity for the lipid membrane (−4.29 kcal/mol) and a logPerm of 1.21, while citric acid had the lowest affinity (−1.40 kcal/mol) and a logPerm of −10.45 (illustrated in Table 1).

The above studies show that 4-HNE stimulates the TRPA1 channel by covalently binding to intracellular cysteine residues.

### 2.4. PDAC Cell Migration Is Impeded by 4-HNE and Recovered in the Presence of the TRPA1 Antagonist

To determine the implication of the activated TRPA1 channel in cell migration, we compared wound area changes in the presence of AITC and 4-HNE and compared them with the control, untreated cells. We found that, after 12 h, PANC-1 cells migrated and covered 45.9 ± 10.9% and 84.1 ± 11.8% of the wound area after 12 and 24 h, respectively (Figure 5, left panel). 

When 50 μM 4-HNE was added into the medium after 12 h, we measured the wound closure by 40.2 ± 12.0%. The wound was covered by 79.7 ± 15.9% after 24 h. 

The statistical analysis of three different experiments, each performed in triplicate, showed that 4-HNE evoked a small but significant inhibition (*p* = 0.01, two-sample Student’s *t*-test, *n* = 3) of cell migration (green box charts, Figure 5, right panel) that was abolished by blocking TRPA1 with 10 μM A-967079 (red box charts, Figure 5, right panel). Instead, adding 10 μM AITC had a much stronger effect after 12 h of treatment, with the wound covered by 34.2 ± 14.3% (*n* = 3), as revealed by parallel experiments (blue chart boxes, Figure 5, right panel).

### 2.5. 4-HNE Stimulates Pre-G1 Phase and Arrests the Cells in the G2/M Phase

Next, we quantified the effect of 4-HNE on PANC-1 cell cycle progression, designating its phases as growth (G0/G1), synthesis (S), mitosis (G2/M), and pre-G1. Our previous results showed that upon activation with AITC, TRPA1 significantly increased the percentage of cells in pre-G1 and decreased that of cells in G0/G1 [10]. We found the same results with the setup used in this study (Figure 6a), showing that in the pre-G2 phase, the percentage of cells augmented to 2.4 ± 0.8 when treated with 50 μM AITC for 24 h, from 0.89 ± 0.1 in control conditions (Figure 6b, red column, *p* = 0.01, Student’s *t*-test). In the same line, the percentage of cells in G2/G1 decreased from 49.5 ± 5.1 in control conditions to 37.6 ± 4.2 in AITC-treated cells (*p* = 0.02, Student’s *t*-test, *n* = 3).

The results in Figure 6c illustrate a typical experiment showing the effect of 50 μM 4-HNE applied for 24 h in the growth medium. We noticed a small but significant decrease in the percentage of cells in G0/G1 (*p* = 0.01, two sample Student’s *t*-test) and S (*p* = 0.02, two sample Student’s *t*-test) compensated by a significant augmentation in the pre-G1 phase (*p* = 0.01, two sample Student’s *t*-test, *n* = 3). However, the percentage of cells in G2/M dramatically increased to 51.61 ± 1.65% following the incubation for 24 h with 50 μM of 4-HNE from 29.67 ± 5.8% in the control, untreated cells. The application of 10 μM A-967079 did not change this trend.

## 3. Discussion

In this study, we found that 4-HNE activates TRPA1 channels endogenously expressed in PANC-1 cells by forming a covalent binding with a pocket-like region within the intracellular N-terminal, shaped by the cysteine residues 621, 641, and 665. The activation of TRPA1 is associated with cell migration and, to a lesser extent, with cell cycle progression.

Many papers suggest that TRPM8 [14], TRPV6 [15], or TRPA1 [10] could be either markers of PDAC or specific targets for cancer therapies based on their localization in the cell membrane, making them easily druggable. Also, TRPA1 agonists and antagonists extracted from natural sources could be appealing for adjuvant treatment. However, contrasting results were obtained regarding the physiological roles of these proteins in carcinogenesis or metastasis, and most importantly, the need for a specific endogenous stimulus makes the applications tremendously scarce. Therefore, the last years’ results showing that lipid peroxidation products, especially 4-HNE, can induce ferroptosis in PDAC [16], together with our current results proving that it activates TRPA1, open up new avenues of studies. 

Our results show, for the first time in PDAC, that 4-HNE, within the concentration range found in cancer patients [17], rapidly activates Ca^2+^ uptake in PDAC cells, as previously demonstrated in transfected HEK 293 cells expressing rat TRPA1 cDNA [11,18]. Because 4-HNE may cause Ca^2+^ overload in isolated cells [19] by mechanisms different from TRPA1 activation, we pretreated the cells with the specific TRPA1 inhibitor A-967079, which completely blocked calcium transients. This result suggests that TRPA1 activation is responsible for the entire calcium uptake. 

4-HNE also modulates gene expression, increasing, for instance, the expression of Nrf2 and other genes related to the antioxidant defense mechanisms [20]. To exclude modulation of TRPA1 expression, we monitored TRPA1 trafficking to the plasma membrane. We also aimed to clarify whether TRPA1 is expressed at the cell membrane or has a subcellular localization, as assumed in our previous reports. Our data indicate that, although TRPA1 is largely present at the cell surface, the channel is mainly distributed in the intracellular compartments in a fine network extending through the cytoplasm. The more extensive distribution of TRPA1 in the intracellular compartments may explain the low number of cells responding to extracellular TRPA1 agonists in PDAC cells [10]. 

We did not find a significant effect of 4-HNE on channel trafficking. However, we found a robust activation of the membrane traffic in the presence of 50 μM hydrogen peroxide (H_2_O_2_). In our hands, H_2_O_2_ did not activate Ca^2+^ uptake, as reported for DRG neurons or HEK693 transfected cells [11,21]. Still, confocal microscopy results show increased TRPA1 expression at the cell membrane after at least 12 h of application. We assume these differences reside in the lower concentration we used in our experiments, but further studies are needed to unveil these discrepancies.

To this end, ROS directly activate the TRPA1 channels by redox modification of the free thiol groups. Previous studies have shown that many TRPA1 electrophilic agonists bind covalently to the Cys621 residue from the NH2 terminus of an α-helix and adjacent to Lys620 [22,23,24]. The residue Lys620 interacts with 4-HNE through van der Waals interactions, facilitating the reactivity of Cys621 [25]. The residue Cys665 forms a hydrogen bond with 4-HNE, while Ile623 forms a hydrogen bond with the sulfur atom of Cys621 involved in the covalent bond with 4-HNE. The residues Ile623 and Thr624 create alkyl interactions with 4-HNE, but the Cys641 residue is not engaged in any of these interactions. From these data, we assume that the strongest covalent binding is formed between 4-HNE and the pocket-like region shaped by the cysteine residues 621, 641, and 665, which are also critical for the interaction of TRPA1 with different electrophile irritants [13]. 

In the quest to determine how the extracellular application of 4-HNE activates TRPA1 by binding to intracellularly located sites, we predicted the membrane permeability using a physics-based method. According to the predicted optimized spatial conformations, both cinnamaldehyde (the considered positive control) and 4-HNE are inserted perpendicularly into the membrane and then flipped into the membrane center while in contact with the lipid hydrophobic tails. Moreover, 4-HNE and other lipid peroxidation products, 4-ONE and 4-HHE, had predicted binding affinities and permeation coefficients that favor their potential to diffuse across the lipid bilayer passively. Therefore, 4-HNE and other similar compounds should permeate the cell membrane with relatively low difficulty, considering their lipophilic nature and favorable membrane transfer energy profiles. 

In this way, we have a clear picture of the activation from the extracellular space, which agrees with studies showing the increase of 4-HNE–protein adducts in serum isolated from cancer patients [26]. Nevertheless, 4-HNE impairs cellular death, migration, or proliferation when endogenously generated. Here, we face two contradictory situations. One implies that 4-HNE can spread from the site of origin and change the structure and function of a protein, leading to tumorigenesis [27]. The other hypothesis states that in low concentrations, 4-HNE inhibits malignancy, as shown by data displaying lower levels of 4-HNE in cancer tissues compared to benign tumors [28]. In PDAC, we find mainly the second situation, as 4-HNE was associated with tumor sensitivity to ferroptosis—the iron-dependent form of non-apoptotic cell death—induced by an organic seleno compound [29].

We followed the hypothesis stating that TRPA1 activation by 4-HNE connects to the tumoral process. Our previous studies have shown that the TRPA1 gene encoding the channel as well as the protein is over-expressed in PDAC cells and patient-derived tissues compared to non-tumoral cell lines or tissue adjacent to the tumor [9]. We have also shown that the channels inhibit migration in both opened and inactive states and generate cell cycle arrest in the pre-G1 phase. Surprisingly, in our hands, 4-HNE evoked a slight, significant inhibition on cell migration after 12 h, which recovered when we added the inhibitor A-967079 to the medium. In contrast, AITC, in parallel experiments and previously reported data [10], inhibited the migration after 12 and 24 h of treatment. We may speculate that the effect on migration is unrelated to Ca^2+^ entry into the cell, as both compounds exhibit activation of Ca^2+^ uptake, but to other unspecific processes. We also know that inhibiting the TRPA1 gene with siRNA activates migration, sustaining the hypothesis that other non-gating mechanisms interfere. 

As expected, the inspection of cell cycle progressions shows that 4-HNE has more complex actions than the simple activation of TRPA1. We measured a tremendous increase, by ~40%, of G2/M cells after 24 of treatment. These results indicate that 4-HNE promotes cell cycle arrest in the G2/M phase in PANC-1 cells, as reported for liver-derived hepatocellular carcinoma cell lines [30]. Apparently, cell cycle arrest and TRPA1 activation are not directly linked, but further silencing studies will bring more insights into the process. It should also be noted that 4-HNE induces a significant increase in the pre-G1 as observed for AITC, although to a lesser extent.

Also, for therapeutic application, we must clearly define the process to choose. Either the activation of TRPA1 could be beneficial based on migration decay and cell cycle arrest in the presence of specific TRPA1 agonists, or, on the contrary, we better inhibit the channel as it prompts evasion from apoptosis through resistance to ROS-inducing chemotherapy [31]. In any circumstance, 4-HNE could be a promising agent by arresting PDAC cells in the G2/M phase, stimulating TRPA1, and inducing consecutive Ca^2+^ entry when applied in micromolar concentration ranges. 

Although important in the context of modulation by 4-HNE of the cell migration and cell cycle in pancreatic adenocarcinoma, this study only shows the effects in cells that express TRPA1. Another cell line with no TRPA1 expression or a more physiological model, such as organoids, should be studied in the following analysis. We foresee performing these experiments and combining them with the effect of H_2_O_2_, which was recently proven to be expressed in high levels in pancreatic cancer cells [32]. Also, we must further study whether the effects are preserved in pancreatic cells resistant to drug or radiation therapies and how the intracellular concentrations of 4-HNE and other products can be quantified and modulated. At first, studies on cells with induced gemcitabine-resistance could be a good model.

In conclusion, our study brings insights into the activation of TRPA1 with a by-product of oxidative stress and opens up the possibility of analyzing the effect of 4-HNE and other ROS in chemotherapy-resistant conditions.

## 4. Materials and Methods

### 4.1. In Vitro Cell Culture Model

PANC-1 (ATCC^®^, CRL-1469) pancreatic ductal epithelioid carcinoma cells were employed in this study as an in vitro cell culture model. These cells were grown in Dulbecco’s modified Eagle medium (DMEM), supplemented with 10% FBS and 1% penicillin/streptomycin mixture (10,000 units/mL penicillin and 10 mg/mL streptomycin) in standard conditions of culture (37 °C, humidified atmosphere of 80 RH and 5% CO_2_). 

### 4.2. Intracellular Nonratiometric Calcium Microfluorimetry 

Cells plated on coverslips were incubated in the extracellular solution (ES) containing 0.02% Pluronic F-127 and 2 μM Calcium Green-1 AM (both from Life Technologies GmbH, Carlsbad, CA, USA) for 30 min at 37 °C. After a recovery period of another 30 min, in standard ES, coverslips were mounted in a Teflon chamber (Harvard Apparatus, Holliston, MA, USA) and then positioned on the stage of an Olympus IX70 inverted microscope (Olympus Corp., Shinjuku, Tokyo, Japan). The experiments started after 5 min for the cells to adapt to the working temperature. A CCD camera (4910, Cohu Inc., Poway, CA, USA) recorded the fluorescence changes. The excitation light at 470 nm was emitted by a Dual OptoLED light source (Cairn Research Ltd., Faversham, Kent, UK), controlled by the Axon Imaging Workbench 2.2 software (Axon Instruments, Molecular Devices, San Jose, CA, USA). The same software was also used for image acquisition and analysis. The OriginPro 9 software (OriginLab Corporation, Northampton, MA, USA) was used for further analyses and data visualization. The experiments were performed at 25 °C. 

In all experiments, we used the standard ES containing (in mM) NaCl, 140; KCl, 4; CaCl_2_, 2; MgCl_2_, 1; HEPES, 10; NaOH, 4.54; and glucose, 5 (pH 7.4 at 25 °C). The ES also contained the vehicle used for dissolving the drugs. Working drug solutions were diluted on the day of the experiment from stock solutions: 100 μM AITC in DMSO, 10 μM A-967079 in DMSO, 50 μM 4-HNE in DMEM. All chemicals were from Merck.

### 4.3. Immunofluorescence

PANC-1 cells were seeded onto coverslips and left to adhere for 24 h. Subsequently, they were incubated for 24 h with 50 μM H_2_O_2_ or 50 μM 4-HNE in complete media. Afterward, samples were fixed with 1% paraformaldehyde in PBS (13 mM NaCl, 2.7 mM KCl, 10 mM Na_2_HPO_4_, and 1.8 mM KH_2_PO_4_) for 1 h at room temperature (RT) and incubated for 3 min with 0.2% Triton-X-100 in PBS to induce cell membrane permeabilization. After that, the samples were incubated overnight in a humidified atmosphere with the blocking buffer solution, 2% BSA in PBS, for two hours, and the primary antibodies: mouse anti-TRPA1-sc-376495 (Santa Cruz Biotechnology, Dallas, TX, USA) and rabbit anti-Calnexin-ab22595 (Abcam, Cambridge, UK). The next day, the coverslips were washed and incubated with the corresponding Alexa Flour-coupled secondary antibodies and for actin staining with Alexa Fluor 594 Phalloidin (Thermo Fisher Scientific, Waltham, MA, USA) for 30 min at RT. Samples were extensively washed and subsequently mounted on glass slides. Images were acquired using the Zeiss LSM 700 (63×, 1.4 NA, oil) microscope using the LSM acquisition software ZEN 3.3 (Zeiss, Oberkochen, Germany). Acquired images were processed using the ImageJ software. Colocalization analysis was performed using the ImageJ JACoP plugin (NIH, Bethesda, MD, USA). The images were split into separate channels and used for threshold processing. The total number of fields analyzed is indicated in the figure legends.

### 4.4. Molecular Docking

Molecular docking of 4-HNE to TRPA1 was performed with Autodock and AutoDock Tools4 [33]. We retrieved the coordinates of 4-HNE from PubChem [34], compound ID 5283344. We used the structural model of a TRPA1 subunit from the AlphaFold Protein Structure Database [35] (ID AF-O75762-F1).

To assess the affinity of 4-HNE toward the Cys pocket known for binding covalent ligands (Cys621, Cys641, Cys665), we performed a prospective docking. The method considered a docking grid centered in the Cys pocket, with a grid size of 126 points in all directions and a distance of 0.375 Å between the grid points. We considered the TRPA1 receptor rigid and the ligand flexible during the docking procedure. Molecular docking was performed using the genetic algorithm, with 100 runs involving 2,500,000 evaluations. From the obtained docking poses, we retained the pose in which the carbon atom of 4-HNE involved in the reaction with 4-HNE was close to the sulfur atom of Cys621 [36].

To predict the most favorable orientation of the ligand when covalently bound to Cys621, we performed the covalent docking [37] of 4-HNE into the Cys pocket. The residue Cys621 bound to 4-HNE [12] was modeled as a single Cys residue comprising its atoms and those of 4-HNE. This residue was treated as flexible while the rest of the protein was rigid. The docking grid (60 points in each direction with 0.375 Å spacing) was centered on this residue. The protein was treated as rigid except for the side chain of Cys621 with 4-HNE. The docking program used a genetic algorithm to explore the full range of ligand conformations in 200 runs, with 25,000,000 evaluations per step.

### 4.5. Prediction of Membrane Permeability

The prediction of membrane permeability was conducted for 4-HNE and two other lipid aldehydes, 4-ONE (4-oxo-2-nonenal) and 4-HHE (4-hydroxy-2-hexenal). We simulated the translocation of small molecules across a dioleoyl phosphatidylcholine lipid bilayer. The membrane permeability was predicted using the PerMM (Permeability of Molecules across Membranes) web server [38], a physics-based method simulating molecules’ transfer from water through the lipid bilayer. We used citric acid as a negative control based on the following rationale: the experimental permeability coefficients available in the PerMM database (used for model validation) show that citric acid permeates the cell membrane exclusively through active transport mechanisms (plasma membrane citrate transporter [39]. Also, citric acid has a molecular weight in the same range as 4-HNE, 4-ONE, and 4-HHE. Cinnamaldehyde is a TRPA1 agonist that binds to intracellular reactive cysteines [40], diffuses through the lipid bilayer, and is used as a positive control. We retrieved the 3D structures of investigated compounds from the PubChem database and performed the simulations at 298 K and pH 7.4. Results were recovered as the binding affinity to the lipid bilayer (ΔG, kcal/mol) and logarithmic permeability coefficients (logPerm). Membrane transfer energy profiles (ΔG_transf_) were also generated as a function of distance from the membrane center denominated as Z.

### 4.6. Wound Healing Assay

The analysis of cell migration was performed using the wound healing assay, as described previously [10]. In brief, cells were seeded in 24-well plates at 6 × 10^4^ cells/well density and grown until a confluent monolayer was formed. A 200 μL sterile plastic micropipette tip simulated an in vivo wound by creating a straight-edged, cell-free zone across the cell monolayer in each well. To ensure reproducibility in surface measurement, two parallel lines perpendicular to each scratch were drawn with a fine permanent marker. The cell monolayer was then washed with the basal medium to remove cell debris, and a complete medium with either 50 μM 4-HNE, 4-HNE plus 50 μM AITC, or 4-HNE plus 10 μM A-967079 was added. The migration progress was acquired at various time points by taking sequential digital photographs of the gap using a phase-contrast Olympus microscope (CKX41, 10× objective) with a digital camera incorporated. The same visual field was used throughout the experiment. The wound gap was measured using ImageJ Pro 8 software; wound closure was calculated using the following formula: wound closure (%) = [(Gap T0 − Gap TΔ)/GapT0] × 100% (where T0 is the area of the wound measured immediately after scratching and TΔ is the area of the wound measured h hours after the scratch was performed).

### 4.7. Cell Cycle

PANC-1 cell cycle distribution was analyzed using flow cytometry after 24 h of treatment with 50 μM AITC, 50 μM 4-HNE, and 10 μM A-967079, applied independently or mixed. Untreated viable cells served as a negative control, and 50% DMSO-treated samples were used as a positive control. Briefly, the cells were plated in 24-well plates at an initial density of 104 cells/well and treated the next day with the above solutions for 24 h. At the end of the treatment, cells were harvested from the culture surface, centrifuged, and washed with cold PBS without Ca^2+^ and Mg^2+^. The cells were resuspended in cold PBS supplemented with 2% FBS and fixed in cold ethanol for at least 48 h at −20 °C. Before the flow cytometer analysis, the samples were washed three times in PBS buffer to remove residual ethanol. Then, the cells were resuspended in 200 µL of PBS, stained with 50 µg/mL propidium iodide, treated with 10 µg/mL RNase A solution, and incubated for 60 min at 37 °C. Cell cycle progression of at least 5000 events/sample was acquired in triplicate using a Beckman Coulter CytoFlex flow cytometer. The data files generated were further analyzed for cell cycle distribution using CytExpert v2 Software (Beckman Coulter, Brea, CA, USA).

## Figures and Tables

**Figure 1 pharmaceuticals-17-00344-f001:**
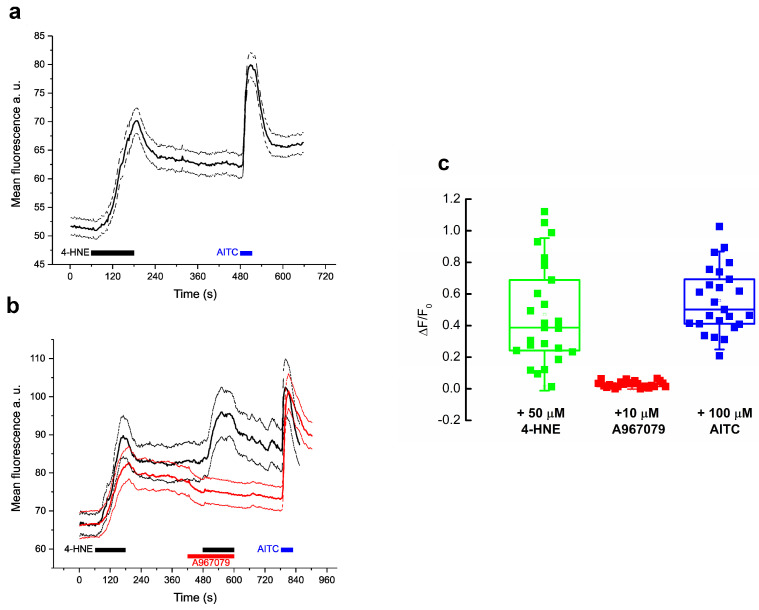
[Ca^2+^]i transients mediated by TRPA1 channels in PANC-1 cells. (**a**) Averages (black line) ± SE (gray lines) of calcium imaging traces of PANC-1 cells treated with 50 μM 4-HNE and 100 μM AITC (*n* = 30). (**b**) Transient responses (averages ± SE) of 4-HNE stimulated cells in control conditions (black traces) and inhibited by 10 μM A-967079 (red traces) (Ctrl, *n* = 17; treated, *n* = 35). (**c**) Normalized mean ΔF/F0 ± SEM of data presented in (**b**). Box charts show the interquartile range (25th to 75th percentiles), the central horizontal line is the median value, and the whiskers represent the range (Ctrl, *n* = 17; treated, *n* = 35).

**Figure 2 pharmaceuticals-17-00344-f002:**
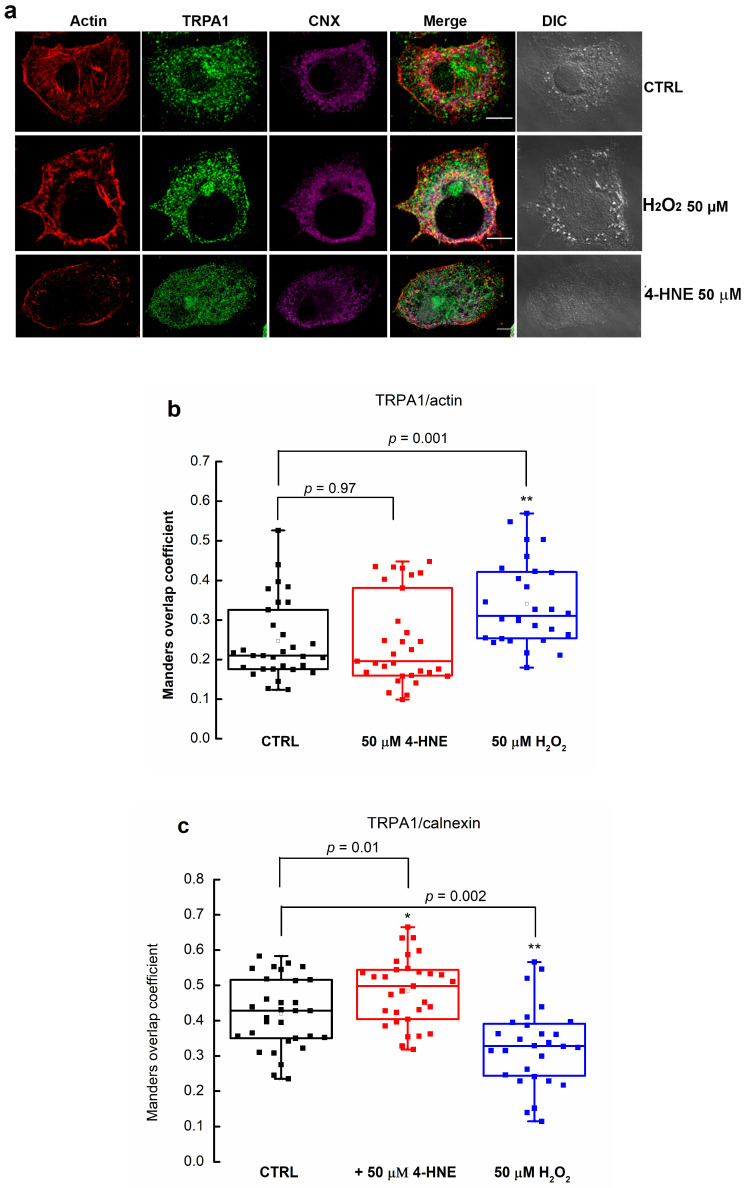
Specific ROS molecules mediate TRPA1 traffic at the plasma membrane. (**a**) Confocal fluorescence microscopy images of PANC-1 cells, non-treated (CTRL) or treated with 50 μM 4-HNE or 50 μM H_2_O_2_ for 24 h, expressing actin (red) partially decorating plasma membrane as cortical actin (used as plasma membrane marker), TRPA1 (green), and calnexin (purple) an ER resident protein (used as ER marker). Scale bars are 10 μm. (**b**,**c**) Quantification of the Manders’ coefficient of (**b**) fraction of plasma membrane actin colocalization with TRPA1 or (**c**) fraction of ER marker-calnexin colocalization with TRPA1. Data represent the raw values of two independent experiments of at least 30 cells. * *p* < 0.05, ** *p* < 0.01.

**Figure 3 pharmaceuticals-17-00344-f003:**
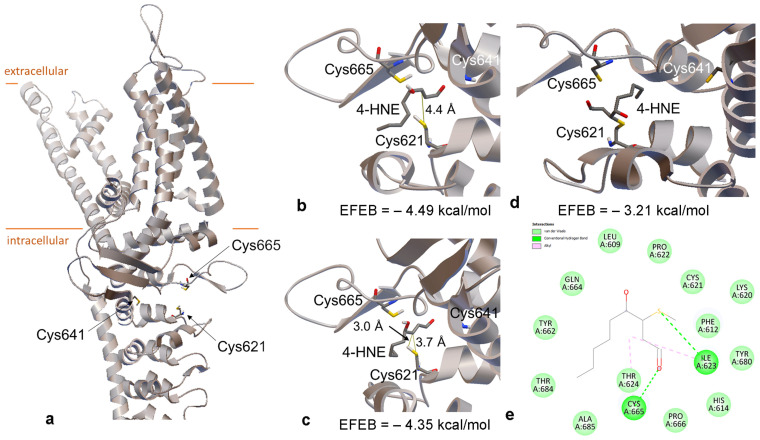
(**a**) Structure of TRPA1 subunit according to the AlphaFold model (ID AF-O75762-F1). The lipid bilayer is represented as orange lines. The three Cys residues from the binding pocket of covalent ligands are labeled in the figure. (**b**) Detail on the best pose of 4-HNE docked into the Cys pocket. The distance between the 4-HNE carbon atom and the Cys621 sulfur atom that will form a covalent bond is labeled in the figure. (**c**) Detail on 4-HNE docked the closest to Cys621. The distances from the 4-HNE carbon atom that will be covalently bound to Cys621 to sulfur and hydrogen atoms from Cys621 are labeled in the figure. (**d**) Detail on the best pose of 4-HNE covalently bound to Cys621. (**e**) A 2D interaction map of covalently bound 4-HNE with the protein. The interacting residues are labeled in the figure and are colored according to the interaction type. In (**b**–**d**), EFEB stands for the estimated free energy of binding calculated for those poses; green dashes represent hydrogen bonds, pink dashes illustrate alkyl interactions.

**Figure 4 pharmaceuticals-17-00344-f004:**
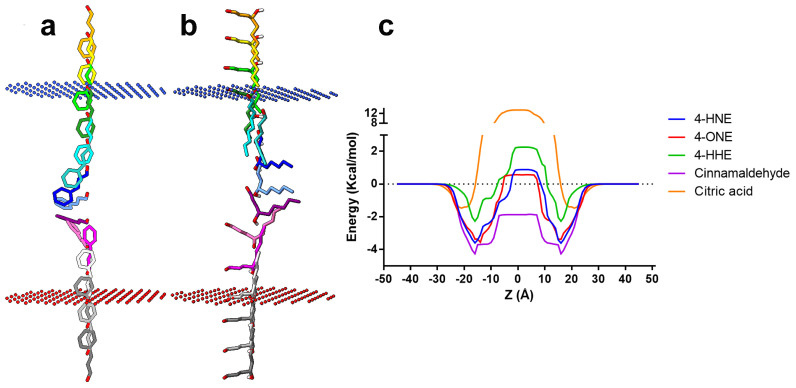
(**a**,**b**) Snapshots illustrating the predicted translocation pathway across the lipid bilayer for positive control (**a**) cinnamaldehyde and (**b**) 4-HNE; (**c**) variation of the transfer energy (ΔG_transf_) as a function of distance from the center of the lipid bilayer (Z).

**Figure 5 pharmaceuticals-17-00344-f005:**
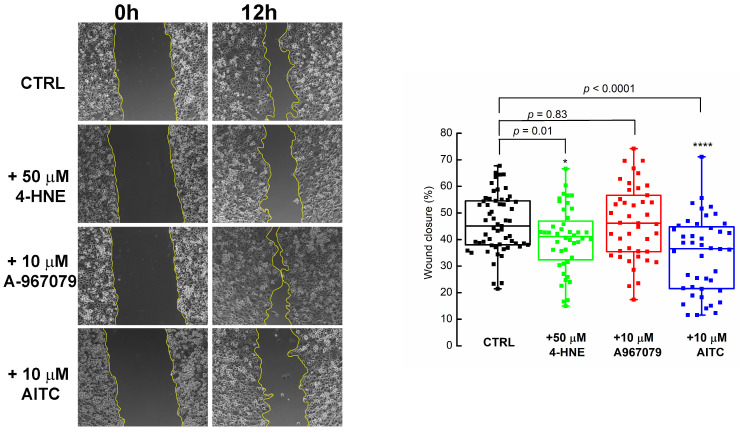
TRPA1 activation inhibited PANC-1 cell migration. Representative wound healing assays (10× magnification) show that 4-HNE slowed cell migration. (**Left**) Representative images of scratched and recovering wounded areas (marked by yellow contours) on confluence monolayers of PANC-1 cells. Time points 0 and 12 h for control (untreated) cells, with 50 μM 4-HNE, when 10 μM A-967079 was added in the medium. (**Right**) Quantification of wound closure using the ImageJ Pro 8 software. Wound closure was determined by measuring the wound area, and at least five pictures/wound were taken for analysis. Data are presented as percentage changes relative to the area of the wounds made in control, untreated cells. Box plots illustrate data from three independent experiments performed in triplicates. * *p* < 0.05, **** *p* < 0.0001.

**Figure 6 pharmaceuticals-17-00344-f006:**
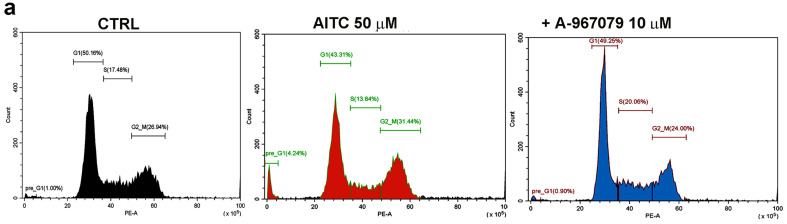
The effect of TRPA1 activators on cell cycle, monitored with flow cytometry. (**a**) AITC decreased the number of cells in pre-G1 and G1 phases. (**b**) Histograms representing the percentage of cell populations in different phases of cell cycles upon challenging the cells with 50 μM AITC and when 10 μM of A-967079 was added in the medium for 24 h. (**c**,**d**) 4-HNE induces cell cycle arrest in the G2/M phase of PANC-1 cells. Representative experiments and histograms showing the percentages of cells in cell cycle phases in the presence of 50 μM 4-HNE alone or in combination with A-967079. Data are mean ± SD from three independent experiments. * *p* < 0.05, ** *p* < 0.01.

**Table 1 pharmaceuticals-17-00344-t001:** Predicted lipid bilayer binding affinity (ΔG) values and permeability coefficients (logPerm).

Compound	ΔG (kcal/mol)	logPerm
4-HNE	−3.62	−0.81
4-ONE	−3.65	−0.67
4-HHE	−2.27	−1.77
Cinnamaldehyde	−4.29	1.21
Citric acid	−1.40	−10.45

## Data Availability

The data presented in this study are available on request from the corresponding author.

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
