# Peer review of "Transient Receptor Potential Ankyrin 1 (TRPA1) Modulation by 4-Hydroxynonenal (4-HNE) in Pancreatic Adenocarcinoma Cell Lines: Putative Roles for Therapies"

_pharmaceuticals, 2024, doi:10.3390/ph17030344_

Round 1

Reviewer 1 Report

Comments and Suggestions for Authors

The presented article is devoted to studying the mechanism of action of 4-hydroxynonenal (4-HNE) through activation of the TRPA1 channel on a specific cell line - pancreatic adenocarcinoma cell. The authors consistently showed the activation of TRPA1 channels, the localization of channels, modeled the molecular interaction of 4-HNE with amino acid residues of the peptide chain of the channel, suggested the mechanism of 4-HNE translocation, and studied the migration and cell cycle of cells under the influence of various agents. The article is written quite clearly and in detail and will be of interest to specialists in this field.
Minor remarks
In my opinion, the caption to Figure 1 needs to be corrected; panels a and b show examples of individual traces - 3 traces each, so there is no need to indicate the number of cells in the caption; the caption to panel b should indicate what color the control cells are shown in and what color the experimental cells are shown in. Panel c represents the processing of the results of data presented in b as follows from the text of the article, and not “of data presented in a and b” as indicated in the signature; here the number of cells for each experiment should be specified in the signature.
Line 86 replace "increased" with "decsreased"
Line 87 -n = 25 what does this mean?

Author Response

We thank the reviewer for the appreciation and suggestions. We performed all the corrections suggested, which are visible in the review mode in the text.

Here are our point-by-point corrections:

In my opinion, the caption to Figure 1 needs to be corrected; panels a and b show examples of individual traces - 3 traces each, so there is no need to indicate the number of cells in the caption; 

We thank the reviewer for the correct remark. In fact, figures a and b present averages ± SE. We corrected the caption accordingly and the corresponding text.

the caption to panel b should indicate what color the control cells are shown in and what color the experimental cells are shown in.

We changed the figure legend and the text.

Panel c represents the processing of the results of data presented in b as follows from the text of the article, and not “of data presented in a and b” as indicated in the signature; here the number of cells for each experiment should be specified in the signature.

Line 86 replace "increased" with "decsreased"

Done  

Line 87 -n = 25 what does this mean?

The number of cells was already mentioned. We corrected.

Reviewer 2 Report

Comments and Suggestions for Authors

The authors of the manuscript entitled “Transient receptor Potential Ankyrin 1 (TRPA1) modulation by 4-hydroxynonenal (4-HNE) in pancreatic adenocarcinoma cell  lines: putative roles for therapies” Florentina Piciu et.al., in this manuscript show for the first time that 4-HNEdifuses the cell membrane and rapidly activates calcium uptake in PDAC cells through TRPA1 activation. Additionally, the authors also show with the Alpha Fold model as 4-HNE forms a covalent binding and stabilizes the open structure of TRPA. In addition, the activation of TRPA1 by 4HNE inhibits cell migration and induces cell cycle arrest in the G2/M phase. Overall, the manuscript is well written with supporting evidence.

There are just some minor suggestions with the study and am curious to know if your study has been carried out on some other pancreatic cell lines and present a comparison across the different cell lines. In addition the authors could also establish the role of 4-HNE and TRPA1 in a more physiological relance in any pancreatic organoid model. It would be ideal if the authors mention about the caveats of their study in the discussion as well as how they could overcome them. The readers would strongly benefit if they could add a paragraph in their discussion.

Author Response

We thank the reviewer for the appreciation and suggestions. We performed all the corrections suggested, which are visible in the review mode in the text.

Here are our point-by-point corrections:

There are just some minor suggestions with the study and am curious to know if your study has been carried out on some other pancreatic cell lines and present a comparison across the different cell lines.

 We have chosen from a batch of different cell lines (Panc-1, MaiPaca-2, BxPC-3, and PCL-12) those with the highest TRPA1 expression. Indeed, the study is relevant for the patients who overexpress the channel.But we intend to continue our study and compare different cell lines with/without TRPA1 and their response to 4-HNE.

In addition the authors could also establish the role of 4-HNE and TRPA1 in a more physiological relance in any pancreatic organoid model. It would be ideal if the authors mention about the caveats of their study in the discussion as well as how they could overcome them. The readers would strongly benefit if they could add a paragraph in their discussion.

 We thank the reviewer for the nice suggestion to pursue our work. We added a paragraph in the discussion section as suggested.

Reviewer 3 Report

Comments and Suggestions for Authors

Thank you very much for giving me an opportunity to review an article from Piciu et al. This is a focused experimental investigation regarding the role of TRPA1 modulation by 4-HNE in PCAC cell line. This study brings a possible novel target of PDAC. The data is clear and very important, then be a fine manuscript for publication.

Minor point:

Authors performed wound healing assays to show the activity of cell migration inhibition of 4-HNE and AITC. Have authors confirmed the effect of 4-HNE or AITC in comparison with gemcitabine, the current gold-standard in the treatment of PDAC?

Comments on the Quality of English Language

None

Author Response

We thank the reviewer for accepting to evaluate our paper, and here is our point-by-point answer:

Authors performed wound healing assays to show the activity of cell migration inhibition of 4-HNE and AITC. Have authors confirmed the effect of 4-HNE or AITC in comparison with gemcitabine, the current gold-standard in the treatment of PDAC?

Gemcitabine induces cell death after 24 of treatment. Therefore, we cannot follow the migration, as the results would be biased by losing cellular viability.  I think a good idea is to follow the results in cells with induced gemcitabine resistance. We added a sentence to explain this future study.